# Evolutionary and Association Analysis of Buffalo FABP Family Genes Reveal Their Potential Role in Milk Performance

**DOI:** 10.3390/genes13040600

**Published:** 2022-03-28

**Authors:** Tingzhu Ye, Aftab Shaukat, Lv Yang, Chao Chen, Yang Zhou, Liguo Yang

**Affiliations:** 1Key Laboratory of Agricultural Animal Genetics, Breeding and Reproduction, College of Animal Science and Technology, Huazhong Agricultural University, Ministry of Education, Wuhan 430070, China; yetingzhu@webmail.hzau.edu.cn (T.Y.); aftabshaukat40@gmail.com (A.S.); yanglv8686@gmail.com (L.Y.); chenchao1995@webmail.hzau.edu.cn (C.C.); yangzhou@mail.hzau.edu.cn (Y.Z.); 2National Center for International Research on Animal Genetics, Breeding and Reproduction (NCIRAGBR), Huazhong Agricultural University, Wuhan 430070, China

**Keywords:** buffalo, FABP, evolutionary, family genes, milk performance

## Abstract

The fatty acid-binding protein (FABP) family gene encode a group of proteins that affect long-chain fatty acid (LCFAs) trafficking and play a crucial function in the regulation of milk fat synthesis. Nevertheless, little is known regarding the identification of members, theevolutionary background, and functional characteristics of FABP genes in buffalo. In this study, in silico analysis was performed to identify the members of FABPs in buffalo. The results revealed that a total of 17 FABP genes were identified. Based on their phylogenetic relationships, these sequences clustered into five groups with similar motif patterns and gene structures. According to positive selection analyses, all duplicated gene pairs containing FABPs in buffalo had Ka/Ks (nonsynonymous/synonymous) ratios that were less than 1, suggesting that they were under purifying selection. Association analysis showed that one SNP in *LOC102401361* was found significantly associated with buffalo milk yield. The expression levels of several FABPs in buffalo mammary epithelial cells were regulated by palmitic and stearic acid treatment. The findings of this study provide valuable information for further research on the role of FABPs in regulating buffalo milk synthesis.

## 1. Introduction

The FABP family is a group of intracellular carriers of bioactive lipids that affect the trafficking of fatty acids [1,2]. LCFAs have been found in different substructures of cells, including the mitochondria, peroxisome, endoplasmic reticulum, lipid droplets, and nucleus, assisting in fat synthesis, assembly, and storage [3]. There are several small proteins in the FABP family of approximately 15 kDa which transport fatty acids together with intracellular retinol and retinoic acid-binding protein [4]. They belong to the intracellular lipid-binding protein (iLBP) superfamily, which includes subfamilies for cellular retinol-binding proteins (CRBPs) and cellular retinoic acid-binding proteins (CRABPs) [5]. Several multiple genes have appeared in this subfamily, giving rise to many iLBPs including 12 FABPs in vertebrates [6,7,8]. However, some members of this subfamily disappeared in some species, such as teleost fishes, *FABP10*, and *FABP11*, which are only found in nonmammalian vertebrates [9]. A previous study reported that the FABP family consisted of four exons and some are distributed on a single chromosome in mammals such as mice, rats, and humans [10,11]. Structural analysis indicated that this subfamily contained a three-element fingerprint domain shared by three motifs termed FATTYACIDBP 1–3 [12]. These three motifs have a similar structure and provide an internal hollow space that assists as a binding site for hydrophobic ligands [13]. According to phylogenetic analysis, FABPs can be divided into three clusters. Cluster 1 contains *FABP1* and *FABP6*, and cluster 2 includes *FABP3*, *FABP4*, *FABP5*, *FABP7*, *FABP8*, *FABP9* [13], and *FABP12* [1], only *FABP2* belongs to cluster 3.

Since the FABP family genes facilitate fatty acids binding and transfection, they also synthesize fatty acids and milk fat. *FABP3, FABP4,* and *FABP5* were up-regulated during lactation and involved fatty acids trafficking towards milk TAG [14]. It was found that *FABP3* provided stearoyl-CoA (or other substrates such as 16:0 and 18:1) [15] to *SCD*, which then released oleic acid to *FABP4*, which then released fatty acids to other enzymes involved in TAG synthesis. Liang et al. [16] found that *FABP3* can regulate milk fat synthesis by modulating the expression of *SREBP1* and *PPARγ* in cattle mammary epithelial cells. The composition of fatty acids in bovine milk is influenced by polymorphisms in *FABP3* and *FABP4* [17], and genetic polymorphisms of *FABP3* are associated with backfat thickness in Korean native cattle [18]. The *FABP4* gene was also linked to fat depth in beef cattle [19] and fat deposition in Australian cattle longissimus [20]. Moreover, it has been reported that the inhibition of *FABP4* or knockout of it reduces osteoarthritis in mice induced by high-fat diets [21], and it was also found to be a therapeutic target for treating obesity-related cancers [22], indicating that this protein may regulate fat metabolism. The expression levels of *FABP1* and *FABP6* were also found upregulated after parturition, promoting fatty acids uptake and intracellular transport [23]. Evidence has demonstrated that different livestock species have different fat percentages in their meat and milk due to FABPs. However, despite buffalo being the second-largest milk producer worldwide, little is known about FABPs in buffalo.

As proteins bind and traffic with fatty acids, the FABP family genes are also, at the same time, affected by fatty acids. For example, it was reported that conjugated linoleic acid persuaded *K-FABP* and *PPAR-delta* expression in the skin of mice. Retinoic acid induces the expression of the *PA-FABP* (psoriasis-associated fatty acid-binding protein) gene in human skin. *H-FABP* expression was also up-regulated by several LCFAs treatments in cultured rat muscle cells [24] and other physiological experiments [25,26]. Long-chain polyunsaturated fatty acids stimulate cellular fatty acid uptake by regulating the expression of *FABP3* and other enzymes in human placental choriocarcinoma cells [27] and in bovine mammary epithelial cells [28].

Buffalo milk is renowned for its high milk fat content and quality [29]. However, as important genes related to milk fat, the buffalo FABP family genes have not been completely identified and analyzed. The completion of the buffalo genome sequence has made it possible to perform genome-wide identification and phylogenetic analysis [30]. Therefore, the current study was designed to identify the FABP family genes in the buffalo genome and analyze their classification, protein motifs, and feature structures. Furthermore, to recognize crucial markers affecting buffalo milk, we performed an association analysis of the FABP family genes with buffalo milk production traits. Next, the expression profile of the subfamily was measured after adding three kinds of LCFAs into buffalo mammary epithelial cells and hepatocytes. Our research provides future functional characterization and influence on milk production of the FABP family genes in buffalo, which is essential for buffalo dairy herd improvement.

## 2. Methods and Materials

### 2.1. Genome-Wide Identification of FABP Genes

Data resources of the genome, proteome, and annotation of river buffalo (*Bubalus bubalis*, assembly UOA_WB_1) [31] were downloaded from the Genome database of NCBI (https://www.ncbi.nlm.nih.gov/, accessed on 1 September 2021). Both the hidden Markov model (HMM) and Basic Local Alignment Search Tool (BLAST) were used to identify all possible FABPs in the present research [32]. One hundred and nine protein sequences of buffalo (*Bubalus ubalis*, 10), cattle (*Bos Taurus*, 40), goat (*Ovis aries*, 20), horse (*Capra hircus*, 18), and sheep (*Equus caballus*, 21) were obtained from the NCBI database (https://www.ncbi.nlm.nih.gov/, accessed on 1 September 2021) (Appendix A). Using these sequences as queries, potential candidate FABPs were searched via BLASTP with a threshold of e-value = 10^−6^. Moreover, HMM profiles were constructed for detecting FABP homologous sequences in HMMER 3.2 [33] (http://hmmer.org/, ccessed on 1 September 2021) with the default setting. The sequences obtained from both approaches were considered candidate FABP family genes. Then, the sequences were submitted to ExPASy (https://web.expasy.org/compute_pi/, accessed on 1 September 2021) to calculate their molecular weight and isoelectric point.

### 2.2. Phylogenetic Analysis of FABPs in Different Organism

Candidate FABP family genes in the other four main milk-supplying livestock species including cattle (21), sheep (18), goat (18), and horse (23) were identified using the same method as in buffalo. The candidate sequence together with buffalo FABPs, a total of 106 amino acid sequences, were aligned by MUSCLE [34] and a neighbor-joining tree and maximum likelihood tree were assembled in MEGA 7.0 with a bootstrap test implemented with 1000 replications (random seed) [35].

### 2.3. Structural Features Analysis

MEGA 7.0 was used to analyze phylogenetic relationships of all FABP amino acid sequences identified to confirm that they belong to the buffalo FABP family genes in order to cluster the FABP amino acid sequences into different subfamilies [36]. To further access the structural diversity of buffalo FABP family members, the sequences were submitted to the MEME 5.0 website (https://meme-suite.org/meme/, accessed on 1 September 2021) to detect conserved motifs [37] and visualized by TBtools (https://github.com/CJ-Chen/TBtools, accessed on 1 September 2021) [38]. To further analyze the detail and function of motifs from the MEME analysis, motif sequences were submitted to the Pfam database (http://pfam.xfam.org/, accessed on 1 September 2021) to search their Pfam matches [39]. The amino acid sequences of the buffalo FABP family were also submitted to Conversed Domains Database (https://www.ncbi.nlm.nih.gov/cdd, accessed on 1 September 2021) to search conversed domains [39] and visualized by TBtools.

### 2.4. Chromosomal Distribution and Gene Duplication Analysis

Gene duplication events with the default parameters of all the identified FABP family members were detected by the Multiple Collinearity Scan Toolkit (MCScanX) software [40]. Then, a chromosome position map with a duplication relationship was constructed using TBtools with the buffalo annotation GFF3 file [38]. The divergence times of duplicated FABP gene pairs were calculated by divergence time = Ks/2λ [41], where Ks is the synonymous substitution rate and λ = 1.26 × 10^−8^ is the clock-like rate in buffalo [42]. To exhibit the positive selection relationship of each tandem duplicated FABP gene pair in buffalo, their synonymous (Ks) and nonsynonymous (Ka) numbers of substitution analysis were performed using TBtools software [38]. Finally, a collinearity map was constructed to detect orthologous gene pairs in buffalo and cattle.

### 2.5. Association Analysis of SNP and Buffalo Milk Traits

Phenotype resources, including 1424 lactation records of 489 Mediterranean Italian buffalos born between 2000 to 2011 and reared in 4 herds in southern Italy, were reported in our previous study [43]. Single nucleotide polymorphisms (SNPs) within and 2000 base pairs upstream of the buffalo FABP family genes were obtained from the genotyping data conducted at Delta Genomics (Edmonton, AB, Canada) using the 90K Axiom Buffalo SNP Array (Affymetrix/Thermo Fisher Scientific, Santa Clara, CA, USA) [43]. Haploview 4.2 was used to compute predicted heterozygosity (He) and Hardy–Weinberg equilibrium (HWE) of the identified SNPs [44]. By using the PIC CALC software, we calculated the polymorphism information content (PIC) of each locus [45]. Six lactation traits comprising peak milk yield, total milk yield, total protein yield, protein percentage, total fat yield, and fat percentage were adjusted to 270-day records to eliminate the effects of environmental factors. The association between each SNP with six 270-day adjusted buffalo milk production traits was performed by the lme4 R-package with the lmer procedure using the following model: Y_ijkl_ = μ + B_i_ + P_j_ + HYS_k_ + G_l_ + e_ijkl_, where Y_ijkl_ = trait observation, μ = overall mean, B_i_ = random effect of buffalo individual, P_j_ = fixed effect of the jth parity (nine classes, 1–9), HYS_k_ = fixed effect of contemporary group constructed with the effects of herd-season and year, G_l_ = fixed effect of the lth genotype, and e_ijkl_ = random residual [46] For the pairwise comparisons among different levels of fixed effects included in the model, the Bonferroni correction for multiple F-testing was applied to the least square means ± SE of the findings for FABPs genotypes.

### 2.6. Cell Culture and Fatty Acid Treatment

The cell culture and medium preparation to culture buffalo mammary epithelial cells (BuMECs) and bovine mammary epithelial cells (BoMECs) were followed with brief modifications as reference [47]. Cell culture medium was prepared using DMEM/F12 (Hyclone, Logan, UT, USA) supplemented with 10% fetal bovine serum (FBS) (Gibco, Billings, MT, USA), 5 ug/mL bovine insulin (Sigma, Ronkonkoma, NY, USA), 1 µg/mL hydrocortisone (Sigma, Ronkonkoma, NY, USA), 1 µg/mL transferrin (Sigma, Ronkonkoma, NY, USA), 1 µg/mL progesterone (Solarbio, Beijing, China), 10 ng/mL EGF (Proteintech, Rosemont, IL, USA), 100 U/mL penicillin, and 100 mg/mL streptomycin. BuMECs were cultured in an incubator at 37 °C under 5% CO_2_. For fatty acid treatment, BuMECs and BoMECs were seeded at 1 × 10^5^/mL into a 6-well plate 48 h before treatment. Three kinds of fatty acids including palmitic acid, stearic acid, and oleic acid (Sigma, Ronkonkoma, NY, USA) were first dissolved with 20 mg/mL in ethanol and then mixed with complete medium to a final concentration of 25 µM, 50 µM, 75 µM, 100 µM, and 125 µM, respectively. Then, the cells were harvested after 6 h in an incubator.

### 2.7. Isolation and Culture of BuMECs

A similar protocol to that used by Vijay Anand et al. was followed to isolate buffalo mammary gland epithelial cells (BuMECs) [47]. In summary, after slaughter, macerated parenchymal tissues from disease-free buffaloes were harvested and transported to the laboratory in DMEM containing 5 µg/mL of streptomycin, 100 U/m of penicillin, and 50 ng/mL of amphotericin. Uteri were washed three times using PBS and trimmed connective tissue and fat. The tissue was cut into small pieces and digested by 0.05% collagenase (Sigma, Ronkonkoma, NY, USA) and 0.05% Hyaluronidase (Sigma, Ronkonkoma, NY, USA) for 5 h at 37 °C. For further digestion, trypsin-EDTA (0.25%) and Dispase (1%; Proteintech, Rosemont, IL, USA) were added to the digested tissue, which was then incubated for 30 min at 37 °C, and the extract was filtered through a 40-cell strainer (Whb-bio, Shanghai, China). The filtrate was centrifuged at 80× *g* for 5 min and washed thrice by PBS. BuMECs were cultured in a culture plate (Corning, Corning, NY, USA) in a growth medium, as described in [47]. After incubation for 5 days, selective trypsinization steps were used to remove the fibroblast cells and get purified BuMECs. The cells were treated 0.25% trypsin-EDTA (Gibco, Billings, MT, USA) and trypsinization was stopped immediately after incubation for 3 min. The detached fibroblast cells in the supernatant were removed. The purified BuMECs were suspended in a freezing medium constituting 90% FBS (Gibco, Billings, MT, USA) and 10% DMSO (Sigma, Ronkonkoma, NY, USA) for the next experiments.

For LCFAs treatment, a density of 1 × 10^5^ cells/cm^2^ was cultured in 6-well plates (Thermo Fisher, Waltham, MA, USA). The medium was changed after one day containing 0, 25 µM, 50 µM, 75 µM, and 100 µM LACFs including palmitic acid, stearic acid, and oleic acid. Then, the cells were allowed to grow for one day and harvested for RNA extraction.

### 2.8. qRT-PCR Analysis

According to the manufacturer’s instructions, cells were extracted with a total RNA kit. Extracted RNA was evaluated using agarose gel electrophoresis and a Nanodrop 2000 spectrophotometer. A total of 2 μg DNase-treated RNA was used for the cDNA synthesis. The qRT-PCR was conducted on the Bio-Rad CFX Maestro (BIO-RAD, Philadelphia, PA, USA) instrument using the QuantiNova SYBR Green PCR Kit (QIAGEN, Hilden, Germany). The *GAPDH* gene was used as an internal control. The 2^−^^ΔΔ^^CT^ method was used to calculate the expression level [48]. The primers used are listed in Appendix A.

## 3. Results

### 3.1. Genome Identification of FABP Family Members

A total of 26 non-redundant buffalo proteins encoded by 17 genes were identified using the BLAST program and HMMER software (Table 1). These genes include 2 genes encoding the cellular retinoic acid-binding protein (CRABP), 9 genes encoding the FABPs, 4 genes encoding the retinol-binding protein, a gene encoding the myelin P2 protein, and an uncharacterized protein. The amino acid length of FABP protein isoforms ranged from 116 to 348, with the predicted molecular weight from 13.16 kDa to 39.13 kDa. The isoelectric point of the transforms ranged from 4.88 to 5.76.

### 3.2. Structural Features of Buffalo FABP Family Members

A phylogenetic analysis was performed on buffalo FABPs to analyze their motif patterns, gene structures, and conserved domains. The phylogenetic analysis showed that buffalo FABPs can be divided into 5 groups (Figure 1A). Ten motifs were identified in buffalo FABPs by MEME research (Figure 1B). Motifs 1 and 2, which were annotated as the lipocalin/cytosolic fatty-acid binding protein family by Pfam search, were the most common motifs around the family members (Appendix A). The gene structural analysis showed that almost all genes except the uncharacterized gene had a complete genome no longer than 30,000 bp. Meanwhile, the FABPs in the same group had a similar structure and exon number (Figure 1C). Furthermore, the conserved domain analysis showed that most of the family members contain domains related to the FABP superfamily (Figure 1D).

### 3.3. Phylogenetic Relationship Analysis of FABP Protein in Five Mammals

To assess the evolutionary relationship of the FABP protein in buffalo and other mammals, a neighbor-joining tree map was built by MUSCLE alignment [34] using all 106 identified protein sequences (Figure 2). Equivalent to buffalo, the FABP protein can be divided into 5 groups. The first group included the most FABP protein sequences (*n* = 45), while the fourth group contained only one FABP gene, including five protein sequences. Besides, by comparing the evolutionary relationship between different organisms, we found that buffalo had a particularly closer relationship with cattle than with other species.

### 3.4. Chromosomal Distribution and Collinearity Analysis of FABP Genes

Based on the mapping data of buffalo with the duplication events analysis, all identified FABP genes were randomly distributed across 11 chromosomes (Figure 3A). Most of FABP family members had 5 genes distributed very closely on chromosome 15. Meanwhile, except for different isoforms of the same genes, many duplication events occurred in different genes. Interestingly, 3 pairs of tandem duplication genes, including *FABP12-FABP4, FABP4-FABP9*, and *FABP9-MP2P*, were distributed on chromosome 15. In addition, the uncharacterized genes *LOC102401361* and *RBP2* were also tandem duplication pairs. For the tandem duplication events of *FABP12-FABP4, FABP4-FABP9,* and *FABP9-MP2P*, the value of the nonsynonymous substitutions per nonsynonymous site (Ka)/the number of synonymous substitutions per synonymous site (Ks) ratios ranged from 0.104 to 0.209 (Table 2), indicating that the FABP family genes might have experienced intense purifying selective pressure during evolution. For the tandem duplication events of *LOC102401361-RBP2* and *LOC102401361-MP2P*, the Ka/Ks ratios were more than 1, showing that *LOC102401361* might be under positive selection. The divergence time of duplicated FABP pairs ranged from 51.328 to 98.92 Mya.

Because the buffalo FABP genes had a particularly closer relationship with cattle than with other species, we further performed a collinearity analysis between cattle and buffalo. The results indicated that 34,710 collinear gene pairs were identified, which account for 81.95% of the total genes (Figure 3B). Although buffalo have different chromosome numbers than cattle, the syntenic gene blocks covered almost all of their chromosomes. Especially, we found 13 pairs of orthologous genes between buffalo and cattle (Table 3). Our findings suggested that these orthologous genes might have a conserved function between the two species during evolution.

### 3.5. Analyses of Association between Traits Related to Buffalo Milk Production

Based on our previous study on 489 buffalo with 1424 lactation records and 60,387 SNPs after quality control for individuals with a call rate ≥97%., we used genotypic and phenotypic data to identify potential markers or genes influencing milk traits in buffalo FABP genes [43]. According to the genotyping dataset, a total of 7 SNPs within 3 FABP genes were filtered after quality control and used in the present study. The allelic and genotypic frequencies, observed and expected heterozygosity, *p*-value of HWE, and PIC are shown in Table 4.

For each SNP, we executed an association analysis with the six milk production traits, then the AX-85106417 located on *LOC102401361* was found significantly associated with MY270. Moreover, the least square mean of individuals with GG (2700 ± 126 kg) was significantly lower than those with AA (3047 ± 70 kg) and AG (3074 ± 72 kg) (Table 5).

### 3.6. Effect of LCFAs on Expression of FABPs

Here, we found that *CRABP1, CRABP2, FABP3, FABP4, FABP5, FABP7*, and *PMP2* were highly expressed in buffalo mammary epithelial cells. After palmitic acid treatment, *CRABP1, CRABP2, FABP3, FABP4*, and *FABP5* mRNA levels were upregulated dramatically with the increase in concentrations, then the *FABP7* mRNA level increased after adding 25, 50, and 75 μM palmitic acid, and the *MP2P* was almost unaffected. As for stearic acid treatment, *CRABP2, FABP3, FABP4,* and *FABP5* mRNA levels increased dramatically with the increase in concentrations, but the *MP2P* declined dramatically with the increase in concentrations; the mRNA level of *CRABP1* reached the lowest when the concentration was 50 uM. It was also found that stearic acid had little effect on *FABP7*. However, oleic acid treatment had no effect or minimal effect on the expression of FABPs (Figure 4).

## 4. Discussion

Buffalo are important livestock in the agricultural economy because they supply milk, meat, and draught for plowing [49]. However, low milk yield has seriously restricted the development of the buffalo industry [50]. It is known that FABPs bind free fatty acids for transport to different organelles for lipid metabolism [51]. Besides, it has also been found that the FABP protein plays an essential role in bovine milk fat synthesis [14]. FABP family proteins have been well characterized in some nonmammalian vertebrates, such as teleost fishes [9], and chicken [1]. However, as important genes related to the synthesis of milk fat, available information on the FABP family genes in milk-producing livestock species, especially in water buffalo, is still limited. The present study identified 26 FABP family protein sequences encoded by 17 genes based on the complete buffalo genome sequence. The identified FABPs were classified into five groups according to their structural features and evolutionary relationships. Consistent with a previous study of cellular FABPs [5,12], this superfamily also included cellular retinol-binding proteins (CRBPs) and cellular retinoic acid-binding proteins (CRABPs) subfamilies. In addition, our motif and conserved domain analysis also proved the classification. We searched 10 motifs from all identified FABP protein sequences, three of which matched to the lipocalin/cytosolic fatty-acid binding protein family, while most of the protein sequences shared at least one of them. All identified protein sequences contained at least one conserved domain related to fatty acid binding. The results were also supported by a previous study [12]. Moreover, we also identified a gene, myelin P2 protein, in the buffalo annotation file; its protein structure is similar to the conserved structure of FABPs [11], and was therefore considered as FABP8 in several organisms such as humans [12,52] and bovine [53]. Thus, it was reasonable to determine the myelin P2 protein as a member of the FABP family genes.

The phylogenetic analysis of FABP family genes displayed a deep understanding of the evolution among them. From the neighbor-joining tree, it was more clear that all FABP family genes could be classified into five groups which is consistent with the result observed in buffalo. Segmental and tandem duplication were common phenomena providing a possibility for novel gene function acquisition in genome evolution [54,55]. A total of 17 identified buffalo FABP family genes were distributed across 13 chromosomes. Here, we found four tandem duplication pairs, including *LOC102401361/RBP2, FABP9/MP2P, FABP4/FABP9*, and *FABP12/FABP4*. For them, the ratio of three pairs of tandem duplication genes (*FABP9/MP2P, FABP4/FABP9*, and *FABP12/FABP4*) was less than 0.3, which indicated that these genes might experience strong purifying selection pressure.

Moreover, we observed that the value of the Ka/Ks ratio for *LOC102401361/RBP2* was more than 1, indicating that the *LOC102401361* gene might be under positive selection during evolution. It is well known that the ratio of Ka and Ks is usually used to determine whether there is selection pressure acting on the protein-coding gene [56,57,58]. We also found that the divergence time of this duplicated pair occurred at 58.515 Mya. These results suggest that the *LOC102401361* gene has the potential to form a novel function. However, this needs to be confirmed further.

To identify more FABPs that affect buffalo milk production traits, we constructed a mixed linear model to perform SNP-traits association analysis. Many SNPs detected in different FABP genes have been significantly associated with milk traits in dairy cows, such as *FABP4* and *FABP3* [17]. However, the present study did not detect relevant genetic variations within these genes due to the minor SNP density of the genotyping arrays. Consequently, only seven SNPs were found in three FABP members, and only one of them was suggested as a potential marker for screening buffalo milk performance. The identification of a significant association for the AX-85106417 marker in this work could indicate that this gene has a potential influence on buffalo milk yield. However, the limited number of animals analysed in this study will make necessary that further research confirm the potential effect of polymorphism on the *LOC102401361* gene on buffalo milk yield.

It is well known that mammary gland epithelial cells play a vital role in the biosynthesis of milk fat in buffalo. Several FABPs have reported to show a high expression in bovine mammary gland tissue and are remarkably up-regulated during lactation [59]. As reported in previous studies, several FABP members served as functional genes for milk fat percentage by regulating the transportation of fatty acid [17,60]. In the present study, we found high expression of *CRABP1, CRABP3, FABP3, FABP4, FABP5, FABP7*, and *MP2P* in buffalo mammary epithelial cells, and most of them were regulated by palmitic acid and stearic acid treatment but not oleic acid treatment, demonstrating that these genes may be involved in the uptake and metabolism of these two kinds of fatty acids. It has been also reported that sheep FABPs were involved in the cellular uptake and metabolism of LCFAs [61]. For example, other study have shown that the expression of *FABP3* in cow mammary epithelial cells was increased after adding 50 and 75 μM oleic acid, 100 μM stearic acid, and 125 μM palmitic acid [16] and it was also affected by short-chain fatty acids (SCFAs) in goat mammary epithelial cells [62]. Those results are in agreement with the results here reported, which suggest that palmitic acid and stearic acid could up-regulate the expression level of *FABP3* in buffalo mammary gland epithelial cells. Furthermore, the expression of several FABPs was found to be affected by palmitic acid and stearic acid, suggesting that FABPs may regulate the metabolism of milk fat by influencing the uptake of LCFAs.

## 5. Conclusions

A total of 17 FABP genes were identified in buffalo genome. Phylogenetic analysis performed here classified these sequences into five groups, showing similar motifs and gene structures within each group. Our analysis suggested that FABPs underwent purifying selection in buffalo and cattle during the evolutionary process. Only one SNP in *LOC102401361* was found significantly associated with buffalo milk yield due to limited number of animals analysed in this study. *LOC102401361* was also found under positive selection during evolution, indicating that the uncharacterized protein might be involved in milk synthesis. The results described in this study showed that the expression of most FABPs in buffalo mammary epithelial cells was regulated by palmitic and stearic acid treatment. The current study provides preliminary results for further investigation into the potential roles that FABP family genes play in the regulation of buffalo milk synthesis.

## Figures and Tables

**Figure 1 genes-13-00600-f001:**
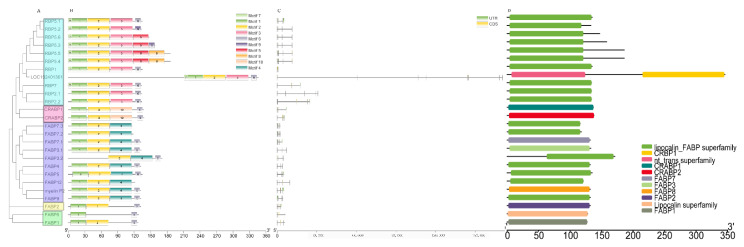
Representations of the recognized FABP proteins and genes isoforms in buffalo. (**A**) The phylogenetic tree was constructed by the N-J method. (**B**) The structure of amino acid sequence boxes represents ten conserved motifs. In the gene structure plot (**C**), the green box, black line, and orange box represent untranslated region (UTR), intron, and coding sequencing (CDS), respectively. Conserved domains (**D**) were searched by NIBI-CCD search.

**Figure 2 genes-13-00600-f002:**
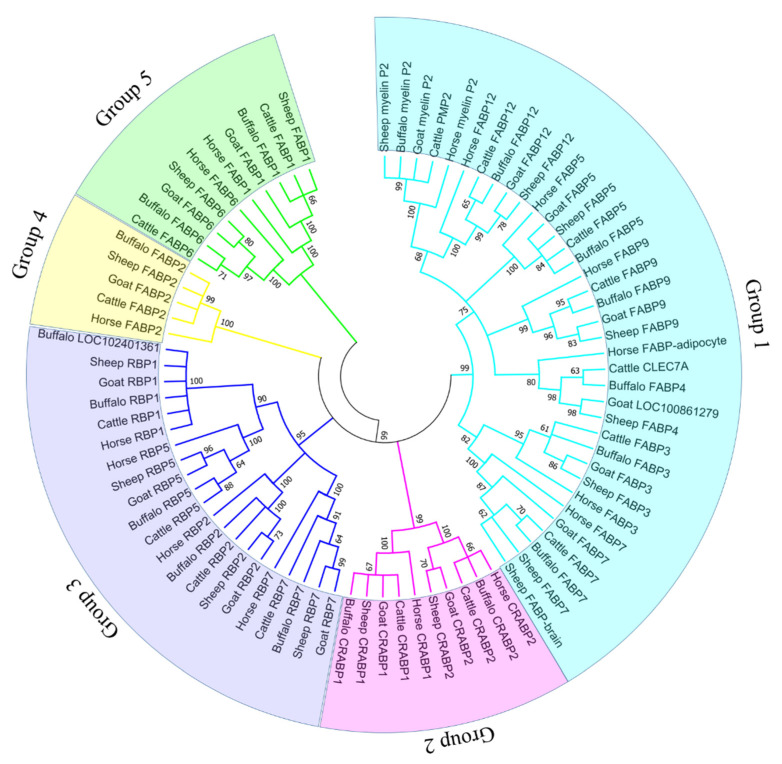
Phylogenetic relationship of FABP proteins in five mammals. Different colored lines indicate different groups. Circles with different colors indicate different groups.

**Figure 3 genes-13-00600-f003:**
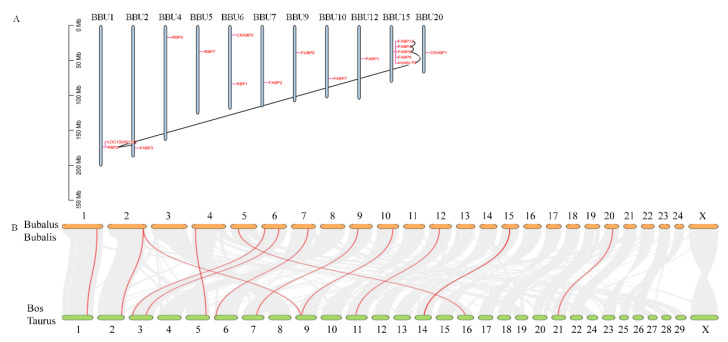
Gene duplication of buffalo FABP genes (**A**) and collinear analysis between buffalo and cattle (**B**)**.** A red line marks the orthologous FABP genes pairs between buffalo and cattle.

**Figure 4 genes-13-00600-f004:**
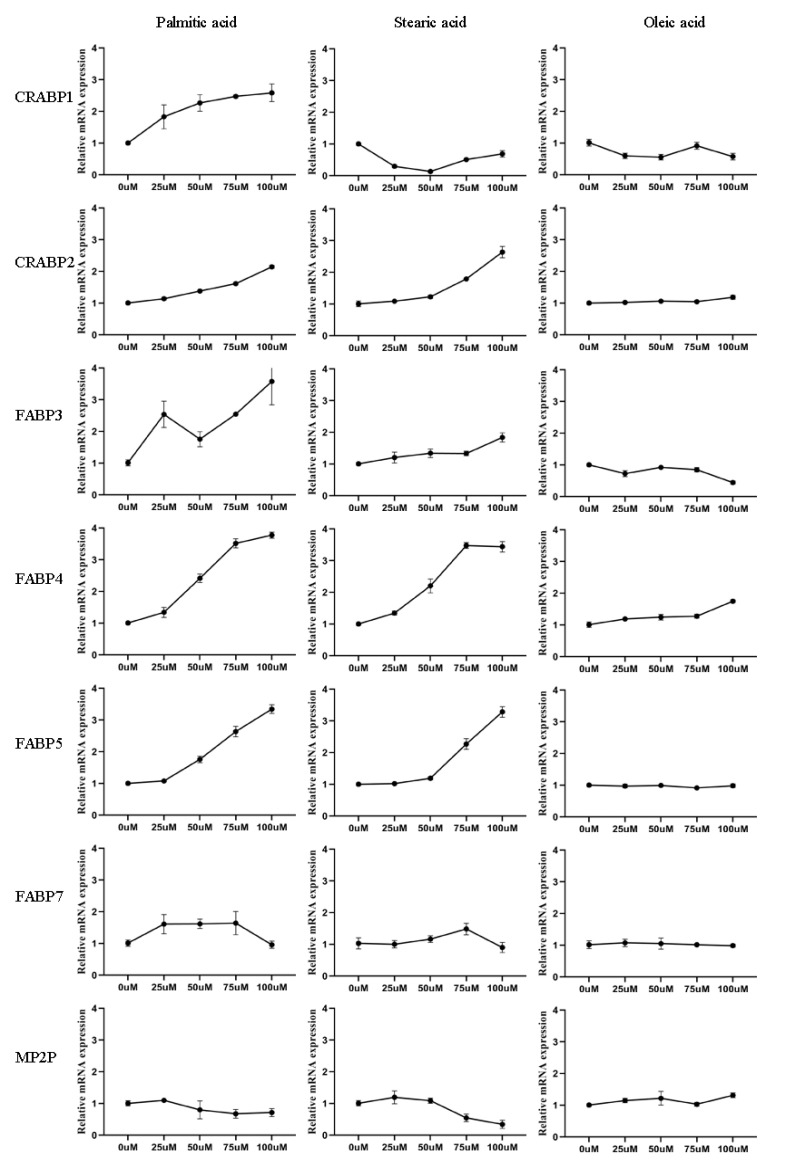
Relative mRNA levels of FABPs in buffalo mammary gland epithelial cells at 24 h after adding palmitic acid, stearic acid, or oleic acid (0, 50, 75, 100, 125, 150 μM), respectively.

**Table 1 genes-13-00600-t001:** Features of the identified FABP protein sequences in buffalo.

List	Protein Isoform	Gene ID	Protein ID	Amino Acids	Isoelectric Point	Mw/kDa	Product
1	*CRABP1*	102392457	XP_006044444.1	137	5.26	15.59	Cellular retinoic acid-binding protein 1
2	*CRABP2*	102406669	XP_006052022.1	138	5.37	15.73	Cellular retinoic acid-binding protein 2
3	*FABP1*	102407733	XP_006074804.1	127	7.78	14.19	Fatty acid-binding protein%2C liver
4	*FABP12*	102410779	XP_006068112.2	121	5.51	13.8	Fatty acid-binding protein 12
5	*FABP2*	102414836	XP_006067721.1	132	5.94	15.09	Fatty acid-binding protein%2C intestinal
6	*FABP3.1*	102394447	NP_001277811.1	133	6.73	14.77	Fatty acid-binding protein%2C heart
7	*FABP3.2*	102394447	XP_025150979.1	171	8.63	18.91	Fatty acid-binding protein%2C heart isoform X1
8	*FABP4*	102410448	NP_001277890.1	132	5.04	14.76	Fatty acid-binding protein%2C adipocyte
9	*FABP5*	102409117	XP_006068107.1	135	7.58	15.07	Fatty acid-binding protein%2C epidermal
10	*FABP6*	102412445	XP_006073509.1	128	6.91	14.37	gastrotropin
11	*FABP7-X1*	102409019	XP_006047648.1	132	5.38	14.95	Fatty acid-binding protein%2C brain isoform X1
12	*FABP7-X2*	102409019	XP_025150573.1	118	5.17	13.42	Fatty acid-binding protein%2C brain isoform X2
13	*FABP7-X3*	102409019	XP_006047649.1	116	5.17	13.16	Fatty acid-binding protein%2C brain isoform X3
14	*FABP9*	102410109	XP_006068110.1	132	9.07	14.87	Fatty acid-binding protein 9
15	*LOC102401361*	102401361	XP_025146712.1	346	6.9	39.13	LOW QUALITY PROTEIN: uncharacterized protein LOC102401361
16	*MP2P*	102409775	XP_006068109.1	132	9.67	14.95	Myelin P2 protein
17	*RBP1*	102389538	XP_006055984.1	135	4.88	15.69	Retinol-binding protein 1
18	*RBP2.1*	102401674	XP_006070028.1	134	5.76	15.7	Retinol-binding protein 2
19	*RBP2.2*	102401674	XP_006070029.1	134	5.76	15.7	Retinol-binding protein 2
20	*RBP5-X1.1*	102393311	XP_025138922.1	186	5.79	21.5	Retinol-binding protein 5 isoform X1
21	*RBP5-X1.2*	102393311	XP_025138923.1	186	5.79	21.5	Retinol-binding protein 5 isoform X1
22	*RBP5-X2*	102393311	XP_025138924.1	158	5.68	18.72	Retinol-binding protein 5 isoform X2
23	*RBP5-X3*	102393311	XP_025138925.1	147	5.29	17.44	Retinol-binding protein 5 isoform X3
24	*RBP5-X4*	102393311	XP_025138926.1	135	5.93	15.96	Retinol-binding protein 5 isoform X4
25	*RBP5-X5*	102393311	XP_025138927.1	133	5.46	15.82	Retinol-binding protein 5 isoform X5
26	*RBP7*	102408740	XP_006076768.1	13c4	6.82	15.54	LOW QUALITY PROTEIN: retinoid-binding protein 7

**Table 2 genes-13-00600-t002:** Ka/Ks ratios estimated in the present work for duplicated FABP genes in buffalo.

Seq_1	Seq_2	Ka	Ks	Ka_Ks	Divergence Time (Mya)
*FABP12*	*FABP4*	0.271232885	1.293454386	0.209696521	51.328
*FABP4*	*FABP9*	0.261352777	2.49279534	0.104843255	98.92
*LOC102401361*	*MP2P*	2.925282375	1.6126178	1.813996085	63.993
*LOC102401361*	*RBP2*	1.99354897	1.474584642	1.351939328	58.515
*MP2P*	*FABP9*	0.234370241	2.248770297	0.104221512	89.237

Note. Ka: nonsynonymous substitution rate; Ks: synonymous substitution rate; Ka_Ks: nonsynonymous/synonymous; Mya: million years ago.

**Table 3 genes-13-00600-t003:** Ka/Ks ratios estimated in the present work for buffalo and cattle orthologous FABP gene pairs.

Buffalo	Cattle	Ka	Ks	Ka_Ks
*LOC102401361*	*RBP1*	NaN	NaN	NaN
*FABP3*	*FABP3*	0.107671646	0.227999676	0.472244734
*FABP3*	*FABP7*	0.31065625	1.679232279	0.184998975
*RBP5*	*RBP5*	0.08500455	0.179602149	0.473293616
*RBP7*	*RBP7*	0.006256553	0.0379828	0.164720704
*CRABP2*	*CRABP2*	0.003127448	0.044008957	0.071063894
*RBP1*	*RBP1*	0.041334963	0.090922001	0.454620033
*FABP2*	*FABP2*	NaN	NaN	NaN
*FABP6*	*FABP6*	0.023564351	0.03739092	0.630215853
*FABP7*	*FABP7*	0	0.049299057	0
*FABP1*	*FABP1*	0.009955898	0.053355825	0.186594405
*FABP5*	*FABP5*	0	0.057832106	0
*FABP12*	*PMP2*	0.257783907	1.849116423	0.139409236
*CRABP1*	*CRABP1*	NaN	NaN	NaN

Note. Ka: nonsynonymous substitution rate; Ks: synonymous substitution rate; Ka_Ks: nonsynonymous/synonymous

**Table 4 genes-13-00600-t004:** Characterization of the SNPs within FABP genes that were genotyped in the Buffalo population of 489 individuals studied in the present work and that passed the quality control filtering.

Gene	Probe Set ID	Location	Genotype	Number	Frequency	Alleles	Rate	Observed He	Predicted He	HWE (*p*-Value)	PIC
*LOC102401361*	AX-85097756	Intro	AA	72	15.7%	A	0.389	0.464	0.475	0.677	0.362
			AG	214	46.5%	G	0.611				
			GG	174	37.8%						
	AX-85049047	Intro	TT	46	10.0%	T	0.307	0.413	0.425	0.616	0.335
			TC	191	41.4%	C	0.693				
			CC	224	48.6%						
	AX-85116471	Intro	AA	127	27.5%	A	0.508	0.465	0.500	0.157	0.375
			AG	215	46.5%	G	0.492				
			GG	120	26.0%						
	AX-85106417	Intro	AA	275	59.5%	A	0.779	0.368	0.344	0.177	0.285
			AG	170	36.8%	G	0.221				
			GG	17	3.7%						
	AX-85072673	Intro	AA	19	4.1%	A	0.183	0.282	0.298	0.306	0.254
			AG	130	28.3%	G	0.817				
			GG	311	67.6%						
*RBP2*	AX-85109932	Intro	TT	258	55.8%	T	0.753	0.390	0.372	0.377	0.303
			TC	180	39.0%	C	0.247				
			CC	24	5.2%						
*RBP5*	AX-85111933	Intro	TT	24	5.2%	T	0.231	0.357	0.355	1.000	0.292
			TC	165	35.7%	C	0.769				
			CC	273	59.1%						

Frequency (%), the frequency of individuals with each haplotype among the population.

**Table 5 genes-13-00600-t005:** SNP association analysis for six milk production traits in buffalo.

Gene	Probe Set ID		Traits (LSM ± SE)					
		Genotype	PM270/kg	MY270/kg	PY270/kg	PP270/%	FY270/kg	FP270/%
*LOC102401361*	AX-85097756	AA	15.8 ± 0.4	3041 ± 82	249 ± 7	8.21 ± 0.13	139 ± 4	4.59 ± 0.04
		AG	15.7 ± 0.3	3040 ± 71	247 ± 6	8.17 ± 0.12	138 ± 3	4.57 ± 0.04
		GG	15.6 ± 0.4	3062 ± 74	248 ± 6	8.16 ± 0.12	139 ± 3	4.58 ± 0.04
		*P*-value	0.666	0.862	0.896	0.867	0.901	0.766
	AX-85049047	CC	15.6 ± 0.3	3048 ± 71	247 ± 6	8.16 ± 0.12	139 ± 3	4.58 ± 0.04
		TC	15.8 ± 0.3	3047 ± 73	249 ± 6	8.2 ± 0.12	138 ± 3	4.56 ± 0.04
		TT	15.8 ± 0.4	3032 ± 88	249 ± 7	8.15 ± 0.14	140 ± 4	4.60 ± 0.04
		*P*-value	0.646	0.97	0.771	0.852	0.852	0.343
	AX-85116471	AA	15.4 ± 0.4	3035 ± 75	245 ± 6	8.16 ± 0.12	138 ± 3	4.59 ± 0.04
		AG	15.8 ± 0.3	3053 ± 72	249 ± 6	8.17 ± 0.12	139 ± 3	4.56 ± 0.04
		GG	15.8 ± 0.4	3045 ± 77	249 ± 6	8.19 ± 0.13	139 ± 3	4.58 ± 0.04
		*P*-value	0.173	0.919	0.57	0.94	0.955	0.387
	AX-85106417	AA	15.6 ± 0.3	3047 ± 70 ^a^	247 ± 6	8.16 ± 0.12	138 ± 3	4.57 ± 0.04
		AG	15.9 ± 0.3	3074 ± 72 ^a^	250 ± 6	8.18 ± 0.12	140 ± 3	4.57 ± 0.04
		GG	15.2 ± 0.6	2700 ± 126 ^b^	230 ± 10	8.23 ± 0.20	130 ± 5	4.61 ± 0.06
		*P*-value	0.197	0.004	0.063	0.913	0.125	0.804
*RBP2*	AX-85072673	AA	15.6 ± 0.5	2997 ± 116	245 ± 9	8.20 ± 0.19	135 ± 5	4.54 ± 0.06
		AG	15.9 ± 0.4	3067 ± 76	249 ± 6	8.17 ± 0.13	139 ± 3	4.57 ± 0.04
		GG	15.6 ± 0.3	3042 ± 70	247 ± 6	8.17 ± 0.12	139 ± 3	4.58 ± 0.03
		*P*-value	0.562	0.743	0.804	0.988	0.725	0.606
*RBP5*	AX-85111933	CC	15.7 ± 0.3	3034 ± 70	247 ± 6	8.17 ± 0.12	138 ± 3	4.57 ± 0.03
		TC	15.7 ± 0.3	3070 ± 74	250 ± 6	8.18 ± 0.12	140 ± 3	4.58 ± 0.04
		TT	15.1 ± 0.5	3015 ± 111	244 ± 9	8.15 ± 0.18	137 ± 5	4.55 ± 0.06
		*P*-value	0.297	0.627	0.44	0.982	0.508	0.79

LSM ± SE represents the least square means ± standard error; PM270, 270-day peak milk yield; MY270, 270-day total milk yield; FY270, 270-day fat yield; FP270, 270-day fat percentage; PY270, 270-day protein yield, and PP270, 270-day protein percentage. Values with different superscripts differ significantly for each SNP each trait at Bonferroni corrected *p* < 0.05.

## Data Availability

Data are contained within the article.

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
