# Peer review of "Evolutionary and Association Analysis of Buffalo FABP Family Genes Reveal Their Potential Role in Milk Performance"

_genes, 2022, doi:10.3390/genes13040600_

Round 1

Reviewer 1 Report

The manuscript entitled “Evolutionary and Association Analysis of Buffalo FABP Gene Family Revealed its Potential Role in Milk Performance” is an interesting article, which mainly aims to investigate the fatty acid-binding protein (FABP) gene family and its potential association with the milk performance and regulation of milk synthesis in buffalo.

Notably, the authors have made a considerable effort to exploit public datasets and perform a bioinformatics analysis using free software. However, lack of innovation seems to be a critical issue associated with this manuscript. The results obtained are not novel enough in a prestigious journal like Genes, except after carrying out significant modifications.

This study has used a medium-density genotyping array (the Affymetrix 90K Axiom Buffalo SNP Array. In addition, the limited number of animals and records analyzed would have reflected on quality of the results obtained. Not surprisingly, only one SNP showed significant association with milk yield.

Conclusion seems as a brief repetition of the results obtained. Could authors retype the conclusion to particularly highlight what is the most utilitarian side of the findings?

Reference list should be updated. No published papers in 2022 were included by the authors as references. More surprisingly, only 2 papers in 2021 and 2 other articles in 2020 were used in this manuscript.

Some notes and comments are listed below:

89-91

Did authors simultaneously consider both HMM and BLAST for selecting candidate FABP family genes?

Please make it clearer to the readers.

Why authors did not include dataset of swamp buffalo in the study?

125-126

Which version of the TBtools software was used for gene duplication analysis?

126-129

Although the ratio between the nonsynonymous substitution rate (Ka) and the synonymous substitution rate (Ks) was computed, the divergence time for each orthologous gene pair in buffalo and cattle was not estimated.

Please calculate it and share the equation used.

146

You wrote, Pj = fixed effect of the jth parity (seven classes, 1–9).

Which 7 parities were included in this research?

In this case, specification of which 7 out of the 9 parities studied is requested.

The quality of Figure 1 is poor. The letters are too small and almost not readable.

Names of the duplicated genes in Figure 3 (A) are not clear enough for reading.  

238-240

“Besides, compared the evolutionary relationship between different organisms, we found that buffalo had a particularly close relationship with buffalo than with other species”

This statement needs to be revised by authors.

Buffalo close to buffalo …… how?!

331-332

“The phylogenetic analysis of FABP family genes in different could display an in-deep 331 insight on the evolutionary among them”

Plz complete the meaning of this statement

Kindly check the attached file.

Reviewer 2 Report

Dear Author, Analysis of  genome wide analysis of FABP gene family in buffalo  is good approach to explore its role in milk performance. 

But there is need

1) to explain the mechanism of effect of long-chain fatty acid on expression of FABPs.

2) What will be the fate of these finding in vivo trial

Reviewer 3 Report

Research paper provides ample information about the role of FABP gene in buffalo. Although similar findings are reported in other bovines but in buffalo this is first report. Paper would be of interest for the related researchers. 

1. Abstract needs a revision for excluding the mention of Genome Wide analysis as it is giving the understanding of doing this experiment in the paper which is not the case as author is referring to his previous published work. This paper is actually an extension of reference 36,38, 9.
2. There should be mention of the in-silico analysis in the abstract
3. There is no mention of fatty acid treatment in the abstract.
4. It is not clear that this family is constituted by how many members? It is better to add a schematic or table in the manuscript indicating the members of the family and subfamily
5. It is mentioned that reports for the gene are already there for cattle, then gene similarity of cattle and buffalo should also be mentioned here
6. Clarification is needed if authors are going to characterize gene family or they are using the existing genomic data for in-silico analysis.
7. Citation of tables and figures in Materials and Methods should be added.
8. Data of SNP genotyping and association has already been published (as mentioned in reference 39) then why is it included in this paper as well?
Regards
